# Covariance shrinkage for autocorrelated data

**Daniel Bartz**
Department of Computer Science
TU Berlin, Berlin, Germany
daniel.bartz@tu-berlin.de

**Klaus-Robert Müller**
TU Berlin, Berlin, Germany
Korea University, Korea, Seoul
klaus-robert.mueller@tu-berlin.de

## Abstract

The accurate estimation of covariance matrices is essential for many signal processing and machine learning algorithms. In high dimensional settings the sample covariance is known to perform poorly, hence regularization strategies such as analytic shrinkage of Ledoit/Wolf are applied. In the standard setting, i.i.d. data is assumed, however, in practice, time series typically exhibit strong autocorrelation structure, which introduces a pronounced estimation bias. Recent work by Sancetta has extended the shrinkage framework beyond i.i.d. data. We contribute in this work by showing that the Sancetta estimator, while being consistent in the high-dimensional limit, suffers from a high bias in finite sample sizes. We propose an alternative estimator, which is (1) unbiased, (2) less sensitive to hyperparameter choice and (3) yields superior performance in simulations on toy data and on a real world data set from an EEG-based Brain-Computer-Interfacing experiment.

## 1 Introduction and Motivation

Covariance matrices are a key ingredient in many algorithms in signal processing, machine learning and statistics. The standard estimator, the sample covariance matrix $\mathbf{S}$, has appealing properties in the limit of large sample sizes $n$: its entries are unbiased and consistent [HTF08]. On the other hand, for sample sizes of the order of the dimensionality $p$ or even smaller, its entries have a high variance and the spectrum has a large systematic error. In particular, large eigenvalues are overestimated and small eigenvalues underestimated, the condition number is large and the matrix difficult to invert [MP67, ER05, BS10]. One way to counteract this issue is to shrink $\mathbf{S}$ towards a biased estimator $\mathbf{T}$ (the shrinkage target) with lower variance [Ste56],

$$\mathbf{C}^{sh} := (1 - \lambda)\mathbf{S} + \lambda\mathbf{T},$$

the default choice being $\mathbf{T} = p^{-1}\text{trace}(\mathbf{S})\mathbf{I}$, the identity multiplied by the average eigenvalue. For the optimal shrinkage intensity $\lambda^\star$, a reduction of the expected mean squared error is guaranteed [LW04]. Model selection for $\lambda$ can be done by cross-validation (CV) with the known drawbacks: for (i) problems with many hyperparameters, (ii) very high-dimensional data sets, or (iii) online settings which need fast responses, CV can become unfeasible and a faster model selection method is required. A popular alternative to CV is Ledoit and Wolf's analytic shrinkage procedure [LW04] and more recent variants [CWEH10, BM13]. Analytic shrinkage directly estimates the shrinkage intensity which minimizes the expected mean squared error of the convex combination with a negligible computational cost, especially for applications which rely on expensive matrix inversions or eigendecompositions in high dimensions.

All of the above algorithms assume i.i.d. data. Real world time series, however, are often non-i.i.d. as they possess pronounced autocorrelation (AC). This makes covariance estimation in high dimensions even harder: the data dependence lowers the *effective sample size* available for constructing the estimator [TZ84]. Thus, stronger regularization $\lambda$ will be needed. In Figure 1 the simple case of an autoregressive model serves as an example for an arbitrary generative model with autocorrelation.

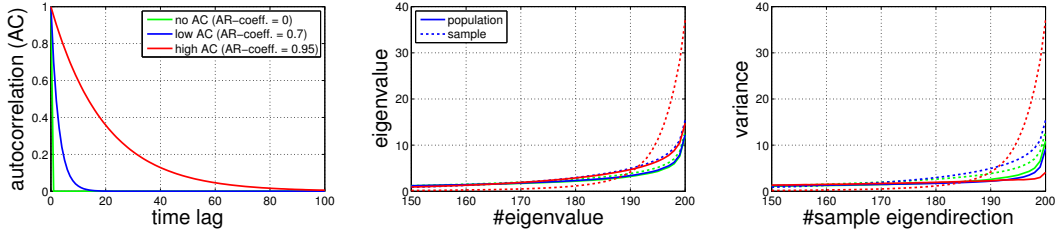

Figure 1: Dependency of the eigendecomposition on autocorrelation. $p = 200$, $n = 250$.

The Figure shows, for three levels of autocorrelation (left), the population and sample eigenvalues (middle): with increasing autocorrelation the sample eigenvalues become more biased. This bias is an optimistic measure for the quality of the covariance estimator: it neglects that population and sample eigenbasis also differ [LW12]. Comparing sample eigenvalues to the *population variance in the sample eigenbasis*, the bias is even larger (right).

In practice, violations of the i.i.d. assumption are often ignored [LG11, SBMK13, GLL+14], although Sancetta proposed a consistent shrinkage estimator under autocorrelation [San08]. In this paper, we contribute by showing in theory, simulations and on real world data, that (i) ignoring autocorrelations for shrinkage leads to large estimation errors and (ii) for finite samples Sancetta's estimator is still substantially biased and highly sensitive to the number of incorporated time lags. We propose a new bias-corrected estimator which (iii) outperforms standard shrinkage and Sancetta's method under the presence of autocorrelation and (iv) is robust to the choice of the lag parameter.

## 2 Shrinkage for autocorrelated data

Ledoit and Wolf derived a formula for the optimal shrinkage intensity [LW04, SS05]:

$$\lambda^{\star} = \frac{\sum_{ij} \text{Var}(S_{ij})}{\sum_{ij} \mathbb{E}\left[(S_{ij} - T_{ij})^2\right]}. \tag{1}$$

The analytic shrinkage estimator $\hat{\lambda}$ is obtained by replacing expectations with sample estimates:

$$\text{Var}(S_{ij}) \quad \longrightarrow \quad \widehat{\text{Var}}(S_{ij}) = \frac{1}{n^2} \sum_{s=1}^{n} \left(x_{is} x_{js} - \frac{1}{n} \sum_{t=1}^{n} x_{it} x_{jt}\right)^2 \tag{2}$$

$$\mathbb{E}\left[(S_{ij} - T_{ij})^2\right] \quad \longrightarrow \quad \widehat{\mathbb{E}}\left[(S_{ij} - T_{ij})^2\right] = (S_{ij} - T_{ij})^2, \tag{3}$$

where $x_{it}$ is the $t^{th}$ observation of variable $i$. While the estimator eq. (3) is unbiased even under a violation of the i.i.d. assumption, the estimator eq. (2) is based on

$$\text{Var}\left(\frac{1}{n} \sum_{t=1}^{n} x_{it} x_{jt}\right) \overset{\text{i.i.d.}}{=} \frac{1}{n} \text{Var}\left(x_{it} x_{jt}\right).$$

If the data are autocorrelated, cross terms cannot be ignored and we obtain

$$\text{Var}\left(\frac{1}{n} \sum_{t=1}^{n} x_{it} x_{jt}\right) = \frac{1}{n^2} \sum_{s,t=1}^{n} \text{Cov}(x_{it} x_{jt}, x_{is} x_{js})$$

$$= \frac{1}{n} \text{Cov}(x_{it} x_{jt}, x_{it} x_{jt}) + \frac{2}{n} \sum_{s=1}^{n-1} \frac{n-s}{n} \text{Cov}(x_{it} x_{jt}, x_{i,t+s} x_{j,t+s})$$

$$=: \frac{1}{n} \Gamma_{ij}(0) + \frac{2}{n} \sum_{s=1}^{n-1} \Gamma_{ij}(s) \tag{4}$$

Figure 2 illustrates the effect of ignoring the cross terms for increasing autocorrelation (larger AR-coefficients, see section 3 for details on the simulation). It compares standard shrinkage to an oracle shrinkage based on the population variance of the sample covariance[1]. The population variance of **S**

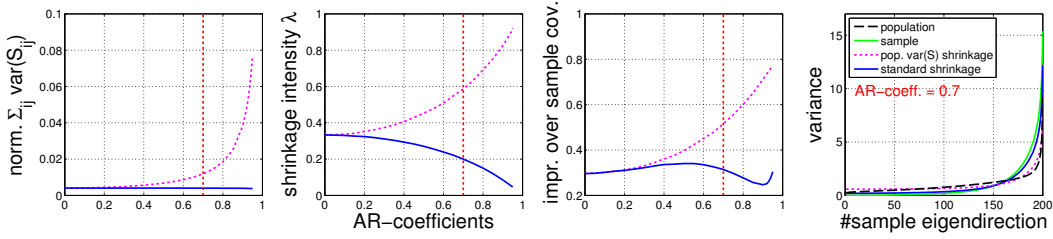

Figure 2: Dependency of shrinkage on autocorrelation. $p = 200$, $n = 250$.

increases because the effective sample size is reduced [TZ84], yet the standard shrinkage variance estimator eq. (2) does not increase (outer left). As a consequence, for oracle shrinkage the shrinkage intensity increases, for the standard shrinkage estimator it even decreases because the denominator in eq. (1) grows (middle left). With increasing autocorrelation, the sample covariance becomes a less precise estimator: for optimal (stronger) shrinkage more improvement becomes possible, yet standard shrinkage does not improve (middle right). Looking at the variance estimates in the sample eigendirections for AR-coefficients of 0.7, we see that the bias of standard shrinkage is only marginally smaller than the bias of the sample covariance, while oracle shrinkage yields a substantial bias reduction (outer right).

**Sancetta-estimator**  An estimator for eq. (4) was proposed by [San08]:

$$\hat{\Gamma}_{ij}^{\text{San}}(s) := \frac{1}{n} \sum_{t=1}^{n-s} \left( x_{it}x_{jt} - S_{ij} \right) \left( x_{i,t+s}x_{j,t+s} - S_{ij} \right), \tag{5}$$

$$\widehat{\text{Var}}\left(S_{ij}\right)^{\text{San,b}} := \frac{1}{n} \left( \hat{\Gamma}_{ij}^{\text{San}}(0) + 2 \sum_{s=1}^{n-1} \kappa(s/b)\hat{\Gamma}_{ij}^{\text{San}}(s) \right), \quad b > 0,$$

where $\kappa$ is a kernel which has to fulfill Assumption B in [And91]. We will restrict our analysis to the truncated kernel $\kappa_{\text{TR}}(x) = \{1 \text{ for } |x| \leq 1, 0 \text{ otherwise}\}$ to obtain less cluttered formulas[2]. The kernel parameter b describes how many time lags are taken into account.

The Sancetta estimator behaves well in the high dimensional limit: the main theoretical result states that for (i) a fixed decay of the autocorrelation, (ii) $b, n \to \infty$ and (iii) $b^2$ increasing at a lower rate than $n$, the estimator is consistent independently of the rate of $p$ (for details, see [San08]). This is in line with the results in [LW04, CWEH10, BM13]: as long as $n$ increases, all of these shrinkage estimators are consistent.

**Bias of the Sancetta-estimator**  In the following we will show that the Sancetta-estimator is suboptimal in finite samples: it has a non-negligible bias. To understand this, consider a lag $s$ large enough to have $\Gamma_{ij}(s) \approx 0$. If we approximate the expectation of the Sancetta-estimator, we see that it is biased downwards:

$$\mathbb{E}\left[\hat{\Gamma}_{ij}^{\text{San}}(s)\right] \approx \mathbb{E}\left[ \frac{1}{n} \sum_{t=1}^{n-s} \left( x_{it}x_{jt}x_{i,t+s}x_{j,t+s} - S_{ij}^2 \right) \right].$$

$$\approx \frac{n-s}{n} \left( \mathbb{E}^2\left[S_{ij}\right] - \mathbb{E}\left[S_{ij}^2\right] \right) = -\frac{n-s}{n}\text{Var}\left(S_{ij}\right) < 0.$$

**Bias-corrected (BC) estimator**  We propose a bias-corrected estimator for the variance of the entries in the sample covariance matrix:

$$\hat{\Gamma}_{ij}^{\text{BC}}(s) := \frac{1}{n} \sum_{t=1}^{n-s} \left( x_{it}x_{jt}x_{i,t+s}x_{j,t+s} - S_{ij}^2 \right), \tag{6}$$

$$\widehat{\text{Var}}\left(S_{ij}\right)^{\text{BC,b}} := \frac{1}{n-1-2b+b(b+1)/n} \left( \hat{\Gamma}_{ij}^{\text{BC}}(0) + 2 \sum_{s=1}^{n-1} \kappa_{\text{TR}}(s/b)\hat{\Gamma}_{ij}^{\text{BC}}(s) \right), \quad b > 0.$$

The estimator $\hat{\Gamma}^{\mathrm{BC}}_{ij}(s)$ is very similar to $\hat{\Gamma}^{\mathrm{San}}_{ij}(s)$, but slightly easier to compute. The main difference is the denominator in $\widehat{\mathrm{Var}}\left(S_{ij}\right)^{\mathrm{BC,b}}$: it is smaller than $n$ and thus corrects the downwards bias.

## 2.1 Theoretical results

It is straightforward to extend the theoretical results on the Sancetta estimator ([San08], see summary above) to our proposed estimator. In the following, to better understand the limitations of the Sancetta estimator, we will provide a complementary theoretical analysis on the behaviour of the estimator for finite $n$.

Our theoretical results are based on the analysis of a sequence of statistical models indexed by $p$. $\mathbf{X}_p$ denotes a $p \times n$ matrix of $n$ observations of $p$ variables with mean zero and covariance matrix $\mathbf{C}_p$. $\mathbf{Y}_p = \mathbf{R}_p^\top \mathbf{X}_p$ denotes the same observations rotated in their eigenbasis, having diagonal covariance $\mathbf{\Lambda}_p = \mathbf{R}_p^\top \mathbf{C}_p \mathbf{R}_p$. Lower case letters $x^p_{it}$ and $y^p_{it}$ denote the entries of $\mathbf{X}_p$ and $\mathbf{Y}_p$, respectively[3]. The analysis is based on the following assumptions:

**Assumption 1** (**A1**, bound on average eighth moment)**.** *There exists a constant $K_1$ independent of p such that*

$$\frac{1}{p} \sum_{i=1}^{p} \mathbb{E}[(x^p_{i1})^8] \leq K_1.$$

**Assumption 2** (**A2**, uncorrelatedness of higher moments)**.** *Let $Q$ denote the set of quadruples $\{i,j,k,l\}$ of distinct integers.*

$$\frac{\sum_{i,j,kl,l \in Q_p} \mathrm{Cov}^2[y^p_{i1} y^p_{j1}, y^p_{k,1+s} y^p_{l,1+s}]}{|Q_p|} = \mathcal{O}\left(p^{-1}\right),$$

*and*

$$\forall s : \frac{\sum_{i,j,kl,l \in Q_p} \mathrm{Cov}\left[(y^p_{i1} y^p_{j1})^2, (y^p_{k,1+s} y^p_{l,1+s})^2\right]}{|Q_p|} = \mathcal{O}\left(p^{-1}\right),$$

*hold.*

**Assumption 3** (**A3**, non-degeneracy)**.** *There exists a constant $K_2$ such that*

$$\frac{1}{p} \sum_{i=1}^{p} \mathbb{E}[(x^p_{i1})^2] \geq K_2.$$

**Assumption 4** (**A4**, moment relation)**.** *There exist constants $\alpha_4$, $\alpha_8$, $\beta_4$ and $\beta_8$ such that*

$$\begin{array}{llll}
\mathbb{E}[y_i^8] & \leq & (1+\alpha_8)\mathbb{E}^2[y_i^4], & \qquad \mathbb{E}[y_i^4] \leq (1+\alpha_4)\mathbb{E}^2[y_i^2], \\
\mathbb{E}[y_i^8] & \geq & (1+\beta_8)\mathbb{E}^2[y_i^4], & \qquad \mathbb{E}[y_i^4] \geq (1+\beta_4)\mathbb{E}^2[y_i^2].
\end{array}$$

**Remarks on the assumptions** A restriction on the eighth moment (assumption **A1**) is necessary because the estimators eq. (2), (3), (5) and (6) contain fourth moments, their variances therefore contain eighths moments. Note that, contrary to the similar assumption in the eigenbasis in [LW04], **A1** poses no restriction on the covariance structure [BM13]. To quantify the effect of averaging over dimensions, assumption **A2** restricts the correlations of higher moments in the eigenbasis. This assumption is trivially fulfilled for Gaussian data, but much weaker (see [LW04]). Assumption **A3** rules out the degenerate case of adding observation channels without any variance and assumption **A4** excludes distributions with arbitrarily heavy tails.

Based on these assumptions, we can analyse the difference between the Sancetta-estimator and our proposed estimator for large $p$:

**Theorem 1** (consistency under "fixed $n$"-asymptotics)**.** *Let **A1**, **A2**, **A3**, **A4** hold. We then have*

$$\frac{1}{p^2} \sum_{ij} \mathrm{Var}\left(S_{ij}\right) = \Theta(1)$$

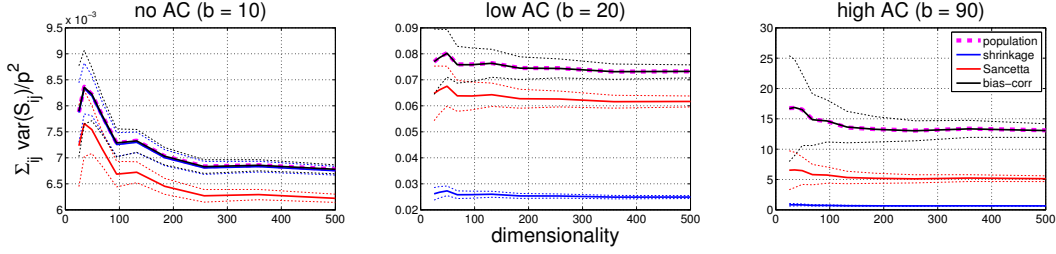

Figure 3: Dependence of the variance estimates on the dimensionality. Averaged over R = 50 models. $n = 250$.

$$\mathbb{E}\left\|\frac{1}{p^2}\sum_{ij}\left(\widehat{\mathrm{Var}}^{San,b}\left(S_{ij}\right) - \mathrm{Var}\left(S_{ij}\right)\right)\right\|^2 = \left(Bias^{San,b} + Bias^{San,b}_{TR}\right)^2 + \mathcal{O}\left(\frac{\sum_j \gamma_j^2}{(\sum_j \gamma_j)^2}\right)$$

$$\mathbb{E}\left\|\frac{1}{p^2}\sum_{ij}\left(\widehat{\mathrm{Var}}^{BC,b}\left(S_{ij}\right) - \mathrm{Var}\left(S_{ij}\right)\right)\right\|^2 = \left(Bias^{BC,b}_{TR}\right)^2 + \mathcal{O}\left(\frac{\sum_j \gamma_j^2}{(\sum_j \gamma_j)^2}\right)$$

*where the $\gamma_i$ denote the eigenvalues of $\mathbf{C}$ and*

$$Bias^{San,b} := -\frac{1}{p^2}\sum_{ij}\left\{\frac{1+2b-b(b+1)/n}{n}\mathrm{Var}\left(S_{ij}\right) - \frac{4}{n^3}\sum_{s=1}^{b}\sum_{t=n-s}^{n}\sum_{u=1}^{n}\mathrm{Cov}\left[x_{it}x_{jt}, x_{iu}x_{ju}\right]\right\}$$

$$Bias^{San,b}_{TR} := -\frac{1}{p^2}\frac{2}{n}\sum_{ij}\sum_{s=b+1}^{n}\frac{n-s}{n}\mathrm{Cov}\left[x_{it}x_{jt}, x_{i,t+s}x_{j,t+s}\right]$$

$$Bias^{BC,b}_{TR} := -\frac{1}{p^2}\frac{2}{n-1-2b+\frac{b(b+1)}{n}}\sum_{ij}\sum_{s=b+1}^{n-1}\mathrm{Cov}\left[x_{it}x_{jt}, x_{i,t+s}x_{j,t+s}\right]$$

*Proof.* see the supplemental material. □

**Remarks on Theorem 1** (i) The mean squared error of both estimators consists of a bias and a variance term. Both estimators have a truncation bias which is a consequence of including only a limited number of time lags into the variance estimation. When $b$ is chosen sufficiently high, this term gets close to zero. (ii) The Sancetta-estimator has an additional bias term which is smaller than zero in each dimension and therefore does not average out. Simulations will show that, as a consequence, the Sancetta-estimator has a strong bias which gets larger with increasing lag parameter $b$. (iii) The variance of both estimators behaves as $\mathcal{O}(\sum_i \gamma_i^2/(\sum_i \gamma_i)^2)$: the more the variance of the data is spread over the eigendirections, the smaller the variance of the estimators. This bound is minimal if the eigenvalues are identical. (iv) Theorem 1 does not make a statement on the relative sizes of the variances of the estimators. Note that the BC estimator mainly differs by a multiplicative factor $> 1$, hence the variance is *larger*, but not relative to the expectation of the estimator.

## 3 Simulations

Our simulations are based on those in [San08]: We average over $R = 50$ multivariate Gaussian AR(1) models

$$\vec{x}_t = \mathbf{A}\vec{x}_{t-1} + \vec{\epsilon}_t,$$

with parameter matrix[4] $\mathbf{A} = \psi_{\mathrm{AC}} \cdot \mathbf{I}$, with $\psi_{\text{no AC}} = 0$, $\psi_{\text{low AC}} = 0.7$, and $\psi_{\text{high AC}} = 0.95$ (see Figure 1). The innovations $\epsilon_{it}$ are Gaussian with variances $\sigma_i^2$ drawn from a log-normal distribution

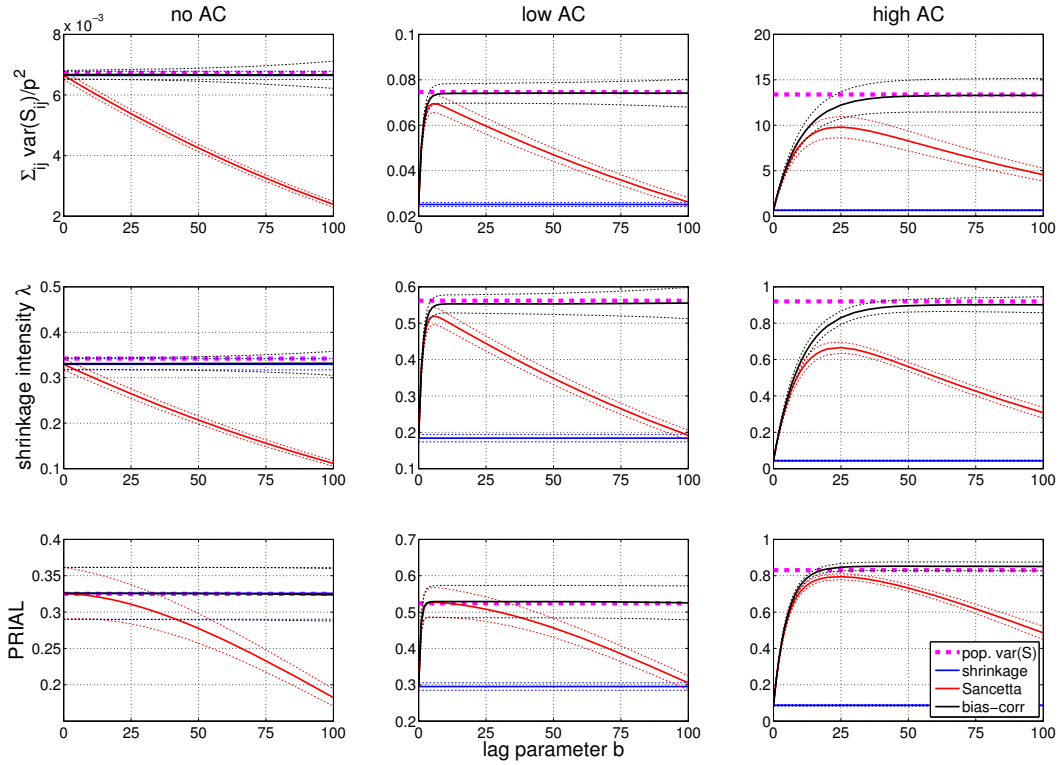

Figure 4: Robustness to the choice of lag parameter $b$. Variance estimates (upper row), shrinkage intensities (middle row) and improvement over sample covariance (lower row). Averaged over R = 50 models. $p = 200$, $n = 250$.

with mean $\mu = 1$ and scale parameter $\sigma = 0.5$. For each model, we generate $K = 50$ data sets to calculate the std. deviations of the estimators and to obtain an approximation of $p^{-2} \sum_{ij} \mathrm{Var}(S_{ij})$.

Simulation 1 analyses the dependence of the estimators on the dimensionality of the data. The number of observations is fixed at $n = 250$ and the lag parameter $b$ chosen by hand such that the whole autocorrelation is covered[5]: $b_{\text{no AC}} = 10$, $b_{\text{low AC}} = 20$ and $b_{\text{high AC}} = 90$. Figure 3 shows that the standard shrinkage estimator is unbiased and has low variance in the *no AC*-setting, but under the presence of autocorrelation it strongly underestimates the variance. As predicted by Theorem 1, the Sancetta estimator is also biased; its remains stays constant for increasing dimensionality. Our proposed estimator has no visible bias. For increasing dimensionality the variances of all estimators decrease. Relative to the average estimate, there is no visible difference between the standard deviations of the Sancetta and the BC estimator.

Simulation 2 analyses the dependency on the lag parameter $b$ for fixed dimensionality $p = 200$ and number of observations $n = 250$. In addition to variance estimates, Figure 4 reports shrinkage intensities and the *percentage improvement in absolute loss* (PRIAL) over the sample covariance matrix:

$$\mathrm{PRIAL}\left(\mathbf{C}^{\{\text{pop., shr, San., BC}\}}\right) = \frac{\mathbb{E}\|\mathbf{S} - \mathbf{C}\| - \mathbb{E}\|\mathbf{C}^{\{\text{pop., shr, San., BC}\}} - \mathbf{C}\|}{\mathbb{E}\|\mathbf{S} - \mathbf{C}\|}.$$

The three quantities show very similar behaviour. Standard shrinkage performs well in the *no AC*-case, but is strongly biased in the autocorrelated settings. The Sancetta estimator is very sensitive to the choice of the lag parameter $b$. For low AC, the bias at the optimal $b$ is small: only a small number of biased terms are included. For high AC the optimal $b$ is larger, the higher number of biased terms causes a larger bias. The BC-estimator is very robust: it performs well for all $b$ large enough to capture the autocorrelation. For very large $b$ its variance increases slightly, but this has practically

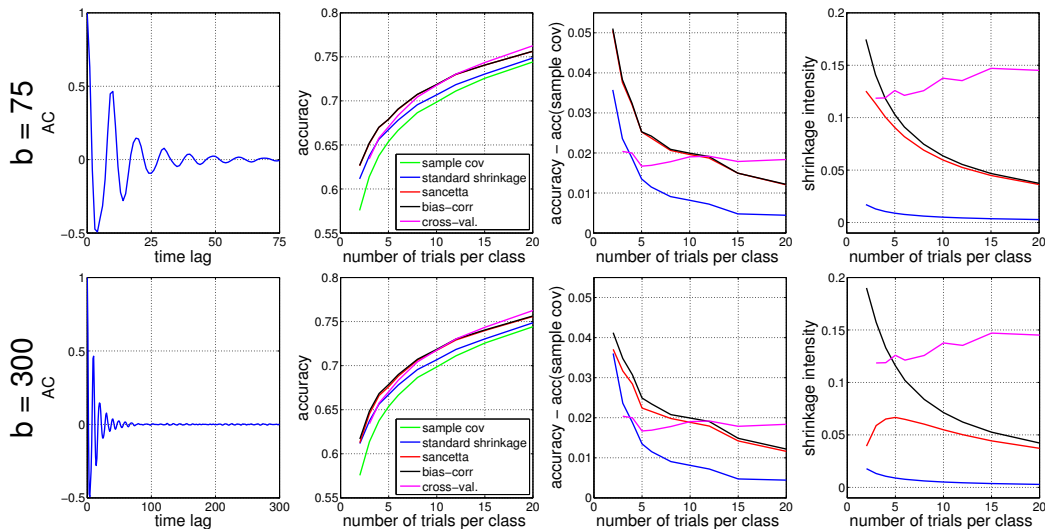

Figure 5: BCI motor imagery data for lag parameter $b = 75$ (upper row) and $b = 300$ (lower row). Averaged over subjects and $K = 100$ runs.

no effect on the PRIAL. An interesting aspect is that our proposed estimator even outperforms shrinkage based on the the population $\text{Var}(S_{ij})$ (calculated by resampling). This results from the correlation of the estimator $\widehat{\text{Var}}(S_{ij})^{\text{BC,b}}$ with the sample estimate eq. (3) of the denominator in eq. (1).

## 4 Real World Data: Brain Computer Interface based on Motor Imagery

As an example of autocorrelated data we reanalyzed a data set from a motor imagery experiment. In the experiment, brain activity for two different imagined movements was measured via EEG ($p = 55$ channels, 80 subjects, 150 trials per subject, each trial with $n_{\text{trial}} = 390$ measurements [BSH$^+$10]). The frequency band was optimized for each subject and from the class-wise covariance matrices, 1-3 filters per class were extracted by Common Spatial Patterns (CSP), adaptively chosen by a heuristic (see [BTL$^+$08]). We trained Linear Discriminant Analysis on log-variance features.

To improve the estimate of the class covariances on these highly autocorrelated data, standard shrinkage, Sancetta shrinkage, cross-validation and and our proposed BC shrinkage estimator were applied. The covariance structure is far from diagonal, therefore, for each subject, we used the average of the class covariances of the other subjects as shrinkage target [BLT$^+$11]. Shrinkage is dominated by the influence of high-variance directions [BM13], which are pronounced in this data set. To reduce this effect, we rescaled, only for the calculation of the shrinkage intensities, the first five principal components to have the same variance as the sixth principal component.

We analyse the dependency of the four algorithms on the number of supplied training trials. Figure 5 (upper row) shows results for an optimized time lag ($b = 75$) which captures well the autocorrelation of the data (outer left). Taking the autocorrelation into account makes a clear difference (middle left/right): while standard shrinkage outperforms the sample covariance, it is clearly outperformed by the autocorrelation-adjusted approaches. The Sancetta-estimator is slightly worse than our proposed estimator. The shrinkage intensities (outer right) are extremely low for standard shrinkage and the negative bias of the Sancetta-estimator shows clearly for small numbers of training trials. Figure 5 (lower row) shows results for a too large time lag ($b = 300$). The performance of the Sancetta-estimator strongly degrades as its shrinkage intensities get smaller, while our proposed estimator is robust to the choice of $b$, only for the smallest number of trials we observe a small degradation in performance. Figure 6 (left) compares our bias-corrected estimator to the four other approaches for 10 training trials: it significantly outperforms standard shrinkage and Sancetta shrinkage for both the larger ($b = 300$, $p \leq 0.01$) and the smaller time lag ($b = 75$, $p \leq 0.05$).

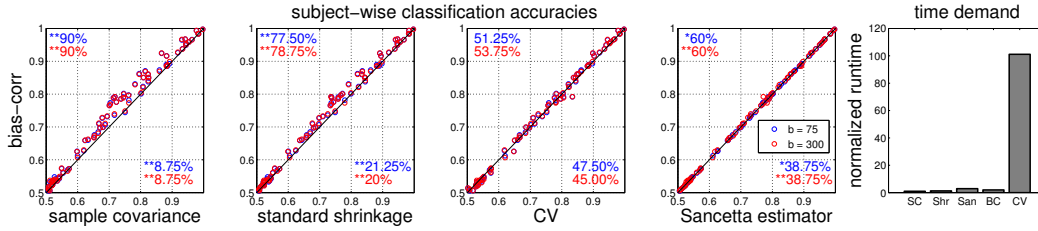

Figure 6: Subject-wise BCI classification accuracies for 10 training trials (left) and time demands (right). $**/* :=$ significant at $p \leq 0.01$ or $p \leq 0.05$, respectively.

Analytic shrinkage procedures optimize only the mean squared error of the covariance matrix, while cross-validation directly optimizes the classification performance. Yet, Figure 5 (middle) shows that for small numbers of training trials our proposed estimator outperforms CV, although the difference is not significant (see Fig. 6). For larger numbers of training trials CV performs better. This shows that the MSE is not a very good proxy for classification accuracies in the context of CSP: for optimal MSE, shrinkage intensities decrease with increasing number of observations. CV shrinkage intensities instead stay on a constant level between 0.1 and 0.15. Figure 6 (right) shows that the three shrinkage approaches ($b = 300$) have a huge performance advantage over cross-validation (10 folds/10 parameter candidates) with respect to runtime.

## 5   Discussion

Analytic Shrinkage estimators are highly useful tools for covariance matrix estimation in time-critical or computationally expensive applications: no time-consuming cross-validation procedure is required. In addition, it has been observed that in some applications, cross-validation is not a good predictor for out-of-sample performance [LG11, BKT+07]. Its speed and good performance has made analytic shrinkage widely used: it is, for example, state-of-the-art in ERP experiments [BLT+11]. While standard shrinkage assumes i.i.d. data, many real world data sets, for example from video, audio, finance, biomedical engineering or energy systems clearly violate this assumption as strong autocorrelation is present. Intuitively this means that the information content per data point becomes lower, and thus the covariance estimation problem becomes harder: the dimensionality remains unchanged but the effective number of samples available decreases. Thus stronger regularization is required and standard analytic shrinkage [LW04] needs to be corrected.

Sancetta already moved the first step into this important direction by providing a *consistent* estimator under i.i.d. violations [San08]. In this work we analysed finite sample sizes and showed that (i) even apart from truncation bias —which results from including a limited number of time lags— Sancetta's estimator is biased, (ii) this bias is only negligible if the autocorrelation decays fast compared to the length of the time series and (iii) the Sancetta estimator is very sensitive to the choice of lag parameter.

We proposed an alternative estimator which is (i) both *consistent* and —apart from truncation bias— *unbiased* and (ii) highly robust to the choice of lag parameter: In simulations on toy and real world data we showed that the proposed estimator yields large improvements for small samples and/or suboptimal lag parameter. Even for optimal lag parameter there is a slight but significant improvement. Analysing data from BCI motor imagery experiments we see that (i) the BCI data set possesses significant autocorrelation, that (ii) this adversely affects CSP based on the sample covariance and standard shrinkage (iii) this effect can be alleviated using our novel estimator, which is shown to (iv) compare favorably to Sancetta's estimator.

### Acknowledgments

This research was also supported by the National Research Foundation grant (No. 2012-005741) funded by the Korean government. We thank Johannes Höhne, Sebastian Bach and Duncan Blythe for valuable discussions and comments.

## Footnotes

[1] calculated by resampling.

[2]in his simulations, Sancetta uses the Bartlett kernel. For fixed $b$, this *increases* the truncation bias.

[3]We shall often drop the sequence index $p$ and the observation index $t$ to improve readability of formulas.

[4]more complex parameter matrices or a different generative model do not pose a problem for the bias-corrected estimator. The simple model was chosen for clarity of presentation.

[5]for $b < 1$, optimal in the *no AC*-setting, Sancetta and BC estimator are equivalent to standard shrinkage.

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
