[Supplementary Material]

# Supplemental Material

# Covariance shrinkage for autocorrelated data

**Daniel Bartz**
Department of Computer Science
TU Berlin, Berlin, Germany
daniel.bartz@tu-berlin.de

**Klaus-Robert Müller**
TU Berlin, Berlin, Germany
Korea University, Korea, Seoul
klaus-robert.mueller@tu-berlin.de

## 1 Theoretical results

Our theoretical results are based on the analysis of a sequence of statistical models indexed by $p$. $\mathbf{X}_p$ denotes a $p \times n$ matrix of $n$ observations of $p$ variables with mean zero and covariance matrix $\mathbf{C}_p$. $\mathbf{Y}_p = \mathbf{R}_p^{\mathrm{T}} \mathbf{X}_p$ denotes the same observations in their eigenbasis, having diagonal covariance $\mathbf{\Lambda}_p = \mathbf{R}_p^{\mathrm{T}} \mathbf{C}_p \mathbf{R}_p$. Lower case letters $x_{it}^p$ and $y_{it}^p$ denote the entries of $\mathbf{X}_p$ and $\mathbf{Y}_p$, respectively[1]. We will restrict the analysis of the Sancetta-estimator to the truncated kernel $\kappa_{\mathrm{TR}}$ to obtain clearer formulas. This *reduces* the bias.

Note that the whole shrinkage framework is invariant to rotations. Switching to the eigenbasis, in which we denote the sample covariance by $\mathbf{S}'$, often simplifies the analysis.

The analysis is based on the following assumptions:

**Assumption 1** (**A1**, bound on average eighth moment)**.** *There exists a constant $K_1$ independent of $p$ such that*

$$\frac{1}{p} \sum_{i=1}^{p} \mathbb{E}[(x_{i1}^p)^8] \leq K_1.$$

**Assumption 2** (**A2**, uncorrelatedness of higher moments)**.** *Let $Q$ denote the set of quadruples $\{i,j,k,l\}$ of distinct integers.*

$$\forall s : \lim_{p \to \infty} \frac{\sum_{i,j,kl,l \in Q_p} \mathrm{Cov}^2[y_{i1}^p y_{j1}^p, y_{k1+s}^p y_{l1+s}^p]}{|Q_p|} = \mathcal{O}\left(p^{-1}\right),$$

*and*

$$\forall s : \lim_{p \to \infty} \frac{\sum_{i,j,kl,l \in Q_p} \mathrm{Cov}\left[(y_{i1}^p y_{j1}^p)^2, (y_{k1+s}^p y_{l1+s}^p)^2\right]}{|Q_p|} = \mathcal{O}\left(p^{-1}\right),$$

*hold.*

**Assumption 3** (**A3**, non-degeneracy)**.** *There exists a constant $K_2$ such that*

$$\frac{1}{p} \sum_{i=1}^{p} \mathbb{E}[(x_{i1}^p)^2] \geq K_2.$$

**Assumption 4** (**A4**, moment relation)**.** *$\exists \alpha_4, \alpha_8, \beta_4$ and $\beta_8$:*

$$
\begin{aligned}
\mathbb{E}[y_i^8] &\leq (1+\alpha_8)\mathbb{E}^2[y_i^4] & \mathbb{E}[y_i^4] &\leq (1+\alpha_4)\mathbb{E}^2[y_i^2] \\
\mathbb{E}[y_i^8] &\geq (1+\beta_8)\mathbb{E}^2[y_i^4] & \mathbb{E}[y_i^4] &\geq (1+\beta_4)\mathbb{E}^2[y_i^2]
\end{aligned}
$$

**Remarks on the assumptions**  A restriction on the eighth moment (assumption **A1**) is necessary because the considered estimators contain fourth moments, their variance therefore is an eighths moment. Note that, contrary to the similar assumption in the eigenbasis in [LW04], **A1** poses no restriction on the covariance structure [BM13].

To quantify how averaging over dimensions occurs, assumption **A2** restricts the correlations of higher moments in the eigenbasis. This assumption is trivially fullfilled for gaussian data, but much weaker (cmp. [LW04]).

Assumption **A3** rules out the degenerate case of adding observation channels without any variance and assumption **A4** excludes distributions with arbitrarily heavy tails.

Based on these assumptions, we can analyse the difference between the Sancetta estimator and our proposed estimator for large $p$:

**Theorem 1** (consistency under "fixed $n$"-asympotics). *Let **A1**, **A2**, **A3**, **A4** hold. We then have*

$$\frac{1}{p^2} \sum_{ij} \mathrm{Var}\,(S_{ij}) = \Theta(1) \tag{1}$$

$$\mathbb{E} \left\| \frac{1}{p^2} \sum_{ij} \left( \widehat{\mathrm{Var}}\,(S_{ij})^{San,b} - \mathrm{Var}\,(S_{ij}) \right) \right\|^2 = \left( Bias^{San,b} + Bias_{TR}^{San,b} \right)^2 + \mathcal{O}\left( \frac{\sum_j \gamma_j^2}{(\sum_j \gamma_j)^2} \right) \tag{2}$$

$$\mathbb{E} \left\| \frac{1}{p^2} \sum_{ij} \left( \widehat{\mathrm{Var}}\,(S_{ij})^{BC,b} - \mathrm{Var}\,(S_{ij}) \right) \right\|^2 = \left( Bias_{TR}^{BC,b} \right)^2 + \mathcal{O}\left( \frac{\sum_j \gamma_j^2}{(\sum_j \gamma_j)^2} \right) \tag{3}$$

*where the $\gamma_i$ denote the eigenvalues of* **C** *and*

$$Bias^{San,b} := -\frac{1}{p^2} \sum_{ij} \left\{ \frac{1 + 2b - b(b+1)/n}{n} \mathrm{Var}\,(S_{ij}) - \frac{4}{n^3} \sum_{s=1}^{b} \sum_{t=n-s}^{n} \sum_{u=1}^{n} \mathrm{Cov}\,[x_{it}x_{jt}, x_{iu}x_{ju}] \right\} \tag{4}$$

$$Bias_{TR}^{San,b} := -\frac{1}{p^2} \frac{2}{n} \sum_{ij} \sum_{s=b+1}^{n} \frac{n-s}{n} \mathrm{Cov}\,[x_{it}x_{jt}, x_{i,t+s}x_{j,t+s}] \tag{5}$$

$$Bias_{TR}^{BC,b} := -\frac{1}{p^2} \frac{2}{n-1-2b+\frac{b(b+1)}{n}} \sum_{ij} \sum_{s=b+1}^{n-1} \mathrm{Cov}\,[x_{it}x_{jt}, x_{i,t+s}x_{j,t+s}] \tag{6}$$

*Proof.* The first statement eq. (1) follows directly from the assumptions (cmp. [BM13]). The error expressions follow from a decomposition into bias and variance.

**Bias of the Sancetta estimator**  Let us first restate the estimator:

$$\hat{\Gamma}_{ij}^{\mathrm{San}}(s) := \frac{1}{n} \sum_{t=1}^{n-s} (x_{it}x_{jt} - S_{ij})(x_{i,t+s}x_{j,t+s} - S_{ij}), \tag{7}$$

$$\widehat{\mathrm{Var}}\left(S_{ij}\right)^{\mathrm{San,b}} := \frac{1}{n} \hat{\Gamma}_{ij}^{\mathrm{San}}(0) + \frac{2}{n} \sum_{s=1}^{n-1} \kappa(s/b) \hat{\Gamma}_{ij}^{\mathrm{San}}(s), \quad b > 0, \tag{8}$$

We start by calculating the bias of the autocovariance estimator eq. (7):

$$\mathbb{E}\left[\hat{\Gamma}_{ij}^{\mathrm{San}}(s)\right] = \mathbb{E}\left[ \frac{1}{n} \sum_{t=1}^{n-s} (x_{it}x_{jt} - S_{ij})(x_{i,t+s}x_{j,t+s} - S_{ij}) \right],$$

$$= \frac{1}{n} \sum_{t=1}^{n-s} \left\{ \mathbb{E}\left[x_{it}x_{jt}x_{i,t+s}x_{j,t+s}\right] - \mathbb{E}\left[x_{it}x_{jt}S_{ij}\right] - \mathbb{E}\left[S_{ij}x_{i,t+s}x_{j,t+s}\right] + \mathbb{E}\left[S_{ij}S_{ij}\right] \right\}$$

$$\mathbb{E}\left[x_{it}x_{jt}S_{ij}\right] = \mathbb{E}\left[x_{it}x_{jt} \frac{1}{n} \sum_{u=1}^{n} x_{iu}x_{ju}\right] = \frac{1}{n} \sum_{u=1}^{n} \mathbb{E}\left[x_{it}x_{jt}x_{iu}x_{ju}\right]$$

$$\mathbb{E}\left[S_{ij}S_{ij}\right] = \mathbb{E}\left[\frac{1}{n}\sum_{v=1}^{n}x_{iv}x_{jv}\frac{1}{n}\sum_{u=1}^{n}x_{iu}x_{ju}\right] = \frac{1}{n^2}\sum_{u,v=1}^{n}\mathbb{E}\left[x_{iv}x_{jv}x_{iu}x_{ju}\right]$$

$$= \frac{n-s}{n}\mathbb{E}\left[x_{it}x_{jt}x_{i,t+s}x_{j,t+s}\right] - \frac{2}{n^2}\sum_{t=1}^{n-s}\sum_{u=1}^{n}\mathbb{E}\left[x_{it}x_{jt}x_{iu}x_{ju}\right] + \frac{n-s}{n^3}\sum_{t,u=1}^{n}\mathbb{E}\left[x_{it}x_{jt}x_{iu}x_{ju}\right]$$

$$= \frac{n-s}{n}\mathrm{Cov}\left[x_{it}x_{jt},x_{i,t+s}x_{j,t+s}\right] - \frac{2}{n^2}\sum_{t=1}^{n-s}\sum_{u=1}^{n}\mathrm{Cov}\left[x_{it}x_{jt},x_{iu}x_{ju}\right] + \frac{n-s}{n^3}\sum_{t,u=1}^{n}\mathrm{Cov}\left[x_{it}x_{jt},x_{iu}x_{ju}\right]$$

$$= \frac{n-s}{n}\mathrm{Cov}\left[x_{it}x_{jt},x_{i,t+s}x_{j,t+s}\right] - \frac{2}{n^2}\sum_{t=1}^{n}\sum_{u=1}^{n}\mathrm{Cov}\left[x_{it}x_{jt},x_{iu}x_{ju}\right] + \frac{2}{n^2}\sum_{t=n-s}^{n}\sum_{u=1}^{n}\mathrm{Cov}\left[x_{it}x_{jt},x_{iu}x_{ju}\right]$$

$$+ \frac{1}{n^2}\sum_{t,u=1}^{n}\mathrm{Cov}\left[x_{it}x_{jt},x_{iu}x_{ju}\right] - \frac{s}{n^3}\sum_{t,u=1}^{n}\mathrm{Cov}\left[x_{it}x_{jt},x_{iu}x_{ju}\right]$$

$$= \frac{n-s}{n}\mathrm{Cov}\left[x_{it}x_{jt},x_{i,t+s}x_{j,t+s}\right] - \frac{n+s}{n}\mathrm{Var}\left[S_{ij}\right] + \frac{2}{n^2}\sum_{t=n-s}^{n}\sum_{u=1}^{n}\mathrm{Cov}\left[x_{it}x_{jt},x_{iu}x_{ju}\right]$$

For the bias of the whole variance estimator eq. (8), we then have

$$\mathbb{E}\left[\widehat{\mathrm{Var}}\left(S_{ij}\right)^{\mathrm{San,b}}\right] = \frac{1}{n}\mathbb{E}\left[\hat{\Gamma}_{ij}^{\mathrm{San}}(0) + 2\sum_{s=1}^{n-1}\kappa(s/b)\hat{\Gamma}_{ij}^{\mathrm{San}}(s)\right]$$

$$= \frac{1}{n}\mathrm{Var}\left[x_{it}x_{jt}\right] + \frac{2}{n}\sum_{s=1}^{n-1}\frac{n-s}{n}\kappa(s/b)\mathrm{Cov}\left[x_{it}x_{jt},x_{i,t+s}x_{j,t+s}\right]$$

$$- \frac{1}{n}\mathrm{Var}\left(S_{ij}\right)\left(1 + 2\sum_{s=1}^{n-1}\frac{n+s}{n}\kappa(s/b)\right)$$

$$+ \frac{4}{n^3}\left(\sum_{s=1}^{n-1}\sum_{t=n-s}^{n}\sum_{u=1}^{n}\mathrm{Cov}\left[x_{it}x_{jt},x_{iu}x_{ju}\right]\kappa(s/b)\right)$$

Plugging in the truncated kernel, we obtain

$$\mathbb{E}\left[\widehat{\mathrm{Var}}\left(S_{ij}\right)^{\mathrm{San,b}}\right] = \frac{1}{n}\mathrm{Var}\left[x_{it}x_{jt}\right] + \frac{2}{n}\sum_{s=1}^{b}\frac{n-s}{n}\mathrm{Cov}\left[x_{it}x_{jt},x_{i,t+s}x_{j,t+s}\right]$$

$$- \frac{1}{n}\mathrm{Var}\left(S_{ij}\right)\left(1 + 2\sum_{s=1}^{b}\frac{n+s}{n}\right)$$

$$+ \frac{4}{n^3}\left(\sum_{s=1}^{b}\sum_{t=n-s}^{n}\sum_{u=1}^{n}\mathrm{Cov}\left[x_{it}x_{jt},x_{iu}x_{ju}\right]\right)$$

$$= \mathrm{Var}\left(S_{ij}\right) - \frac{2}{n}\sum_{s=b+1}^{n}\frac{n-s}{n}\mathrm{Cov}\left[x_{it}x_{jt},x_{i,t+s}x_{j,t+s}\right]$$

$$- \frac{1}{n}\mathrm{Var}\left(S_{ij}\right)\left(1 + 2\sum_{s=1}^{b}\frac{n+s}{n}\right)$$

$$+ \frac{4}{n^3}\left(\sum_{s=1}^{b}\sum_{t=n-s}^{n}\sum_{u=1}^{n}\mathrm{Cov}\left[x_{it}x_{jt},x_{iu}x_{ju}\right]\right)$$

We have shown that eq. (4) and eq. (5) hold.

**Bias of the BC-estimator** Let us again first restate the estimator:

$$\hat{\Gamma}_{ij}^{\mathrm{BC}}(s) := \frac{1}{n}\sum_{t=1}^{n-s}\left(x_{it}x_{jt}x_{i,t+s}x_{j,t+s} - S_{ij}^2\right), \tag{9}$$

$$\widehat{\mathrm{Var}}\left(S_{ij}\right)^{\mathrm{BC,b}} := \frac{1}{n-1-2b+b(b+1)/n}\left(\hat{\Gamma}_{ij}^{\mathrm{BC}}(0) + 2\sum_{s=1}^{n-1}\kappa_{TR}(s/b)\hat{\Gamma}_{ij}^{\mathrm{BC}}(s)\right). \qquad (10)$$

For the bias-corrected estimator, we also start by calculating the bias of the autocovariance estimator eq. (9):

$$
\begin{aligned}
\mathbb{E}\left[\hat{\Gamma}_{ij}^{\mathrm{BC}}(s)\right] &= \mathbb{E}\left[\frac{1}{n}\sum_{t=1}^{n-s}\left(x_{it}x_{jt}x_{i,t+s}x_{j,t+s} - S_{ij}^2\right)\right]\\
&= \mathbb{E}\left[x_{it}x_{jt}x_{i,t+s}x_{j,t+s}\right] - \mathbb{E}\left[S_{ij}^2\right]\\
&= \mathbb{E}\left[x_{it}x_{jt}x_{i,t+s}x_{j,t+s}\right] - \mathrm{Var}\left[S_{ij}\right] - \mathbb{E}^2\left[S_{ij}\right]\\
&= \mathbb{E}\left[x_{it}x_{jt}x_{i,t+s}x_{j,t+s}\right] - \mathrm{Var}\left[S_{ij}\right] - \mathbb{E}^2\left[(x_{it}x_{jt})\right]\\
&= \frac{n-s}{n}\mathrm{Cov}\left[x_{it}x_{jt}, x_{i,t+s}x_{j,t+s}\right] - \frac{n-s}{n}\mathrm{Var}\left[S_{ij}\right]
\end{aligned}
$$

Plugging this into the expression for the whole variance estimator eq. (10), we obtain

$$
\begin{aligned}
\mathbb{E}\left[\widehat{\mathrm{Var}}\left(S_{ij}\right)^{\mathrm{BC,b}}\right] &= \frac{1}{n-1-2b+\frac{b(b+1)}{n}}\mathbb{E}\left[\hat{\Gamma}_{ij}^{\mathrm{BC}}(0) + 2\sum_{s=1}^{n-1}\kappa_{\mathrm{TR}}(s/b)\hat{\Gamma}_{ij}^{\mathrm{BC}}(s)\right]\\
&= \frac{1}{n-1-2b+\frac{b(b+1)}{n}}\left(\mathrm{Var}\left[x_{it}x_{jt}\right] + 2\sum_{s=1}^{b}\frac{n-s}{n}\mathrm{Cov}\left[x_{it}x_{jt}, x_{i,t+s}x_{j,t+s}\right]\right.\\
&\qquad\qquad\left. - (1+2b-\frac{b(b+1)}{n})\mathrm{Var}\left[S_{ij}\right]\right)\\
&= \frac{1}{n-1-2b+\frac{b(b+1)}{n}}\left(n\mathrm{Var}\left[S_{ij}\right] - 2\sum_{s=b+1}^{n-1}\frac{n-s}{n}\mathrm{Cov}\left[x_{it}x_{jt}, x_{i,t+s}x_{j,t+s}\right]\right.\\
&\qquad\qquad\left. - (1+2b-\frac{b(b+1)}{n})\mathrm{Var}\left[S_{ij}\right]\right)\\
&= \mathrm{Var}\left[S_{ij}\right] - \frac{2}{n-1-2b+\frac{b(b+1)}{n}}\sum_{s=b+1}^{n-1}\frac{n-s}{n}\mathrm{Cov}\left[x_{it}x_{jt}, x_{i,t+s}x_{j,t+s}\right].
\end{aligned}
$$

We have shown that eq. (6) holds.

**Bound on the variance** The variance terms are more involved: Let us exemplarily consider the variance of

$$\frac{1}{p^2}\widehat{\mathrm{Var}}\left(S_{ij}'\right)^{\mathrm{BC,b}} = \frac{1}{p^2(n-1-2b)}\sum_{ij}\left(\hat{\Gamma}_{ij}^{\mathrm{BC}}(0) + 2\sum_{s=1}^{n-1}\kappa_{\mathrm{TR}}(s/b)\hat{\Gamma}_{ij}^{\mathrm{BC}}(s)\right),$$

We can simplify the analysis of the variance of this expression by looking at each $\hat{\Gamma}(s)$ separately: there is only a finite number of terms. Derivations are similar for all terms, we here only show the first term:

$$\mathrm{Var}\left(\frac{1}{p^2(n-1-2b)}\sum_{ij}\hat{\Gamma}_{ij}(0)\right) = \mathrm{Var}\left(\frac{1}{p^2(n-1-b)n}\sum_{ij}\left(\sum_{s}y_{is}^2y_{js}^2 - \frac{1}{n}\sum_{ss'}y_{is'}y_{js'}y_{is}y_{js}\right)\right)$$

Ignoring constants we again treat each term separately, the first term yields

$$
\begin{aligned}
\mathrm{Var}\left(\frac{1}{p^2}\sum_{ij}y_{i1}^2y_{j1}^2\right) &= \frac{1}{p^4}\sum_{ijkl}\mathrm{Cov}\left(y_{i1}^2y_{j1}^2, y_{k1}^2y_{l1}^2\right)\\
&= \frac{1}{p^4}\sum_{i,j,i',j'\in Q}\mathrm{Cov}\left(y_{i1}^2y_{j1}^2, y_{k1}^2y_{l1}^2\right) + \frac{1}{p^4}\sum_{i,j,i',j'\in R}\mathrm{Cov}\left(y_{i1}^2y_{j1}^2, y_{k1}^2y_{l1}^2\right).
\end{aligned}
$$

where we decomposed the set of integers into two disjoint subsets: $\{1, \ldots, p\}^4 = Q \cup R$, where $Q$ is the set of distinct integers and $R$ is the remainder. The first term is taken care of by **A4**. For the second term, $(i, i', j, j') \in R$, we have

$$\frac{1}{p^4} \sum_{(i,i',j,j') \in R} \mathrm{Cov}\left(y_{i1}^2 y_{j1}^2, y_{i'1}^2 y_{j'1}^2\right)$$

$$\leq \frac{6}{p^4} \sum_{i,j,i'} \mathrm{Cov}\left(y_{i1}^2 y_{j1}^2, y_{i'1}^2 y_{i1}^2\right) + \mathrm{Cov}\left(y_{i1}^2 y_{j1}^2, y_{i'1}^4\right)$$

$$\leq \frac{6}{p^4} \sum_{i,j,i'} \sqrt{\mathbb{E}[y_{i1}^4 y_{j1}^4]} \sqrt{\mathbb{E}[y_{i'1}^4 y_{i1}^4]} + \sqrt{\mathbb{E}[y_{i1}^4 y_{j1}^4]} \sqrt{\mathbb{E}[y_{i'1}^8]}$$

$$\leq \frac{6}{p^4} \sum_{i,j,i'} \sqrt[4]{\mathbb{E}[y_{i1}^8]\mathbb{E}[y_{j1}^8]} \sqrt[4]{\mathbb{E}[y_{i'1}^8]\mathbb{E}[y_{i1}^8]} + \sqrt[4]{\mathbb{E}[y_{i1}^8]\mathbb{E}[y_{j1}^8]} \sqrt{\mathbb{E}[y_{i'1}^8]}$$

$$\leq \frac{12(1 + \alpha_8)}{p^4} \sum_{i,j,i'} \mathbb{E}[y_{i1}^2]\mathbb{E}[y_{j1}^2]\mathbb{E}^2[y_{i'1}^2]$$

$$= \mathcal{O}\left(\frac{\sum_i \gamma_i^2}{\left(\sum_i \gamma_i\right)^2}\right),$$

where in the last equality we have used that $\sum_{i=1}^p \gamma_i = \Theta(p)$. The same asymptotic behaviour can be derived for the other terms following very similar steps. The bounds in eq. (2) and eq. (3) follow. □

## Footnotes

[1]We shall often drop the sequence index $p$ and the observation index $t$ to improve readability of formulas.