[Reviews · NeurIPS 2014]

Submitted by Assigned_Reviewer_13

840:

this is a very nice paper, with compelling theoretical, simulated, and real data results. i have a few majorish issues, and some minor ones.

major

-- one can choose lambda via CV or some theoretical tool. if the theoretical tool has no parameters, it is a clear win. however, there is a truncation parameter here. this manuscript did not convey to me *how* to choose b, and importantly, the extent to which the results are robust to this choice of b. if this method is to be adopted as the de facto standard, some discussion about how to choose b and robustness to that choice is necessary.

-- given that the main justification of using this method over CV is computational time, one might also acknowledge that practitioners always weigh a trade-off between accuracy and time.
clearly, this method is faster than CV, assuming we have a good way of choosing b. but, how accurate is it? if it is much less accurate, than the improvement in time might not be so useful. for example, in the real data example, we could simply use the average class covariances for the other subjects. this would be fast, parameter free, and maybe just as accurate?

minor

-- in eq 5, b is some constant that satisfies some properties as a function of n? please clarify more formally the assumptions on b. also, please explain b. please define the truncation kernel here.

-- "we will provide a complementary analysis on the behaviour of the estimator for finite n."

perhaps state a 'complementary theoretical analysis', i was led to believe you possibly meant only numerical, which of course, is much weaker.

-- line 206, space missing

-- remarks on thm 1: i would like more explanation of the relative size of the 3 biases. the biases are a function of b, n, s and covariances. some plotting showing the relative magnitude, say, of bias(San) vs bias(BC) would be very helpful. for example, a heatmap showing bias(San)-bias(BC) for fixed n when varying b and s, or fixed function b_n and varying n & s.

-- i don't understand the simulation setting. please explain it more clearly, with equations, the notation for the 'parameter matrix' is unclear to me, what are '/' meant to denote? also, i don't know the abbreviation 'cmp'. if you are just trying to save space, i recommend removing some paragraph breaks, and keep content as clear as possible.

-- a supplementary figure justifying footnote 4 is requested.

-- "We average over R multivariate " ok, what do you set R to be for these simulations?

-- i think a better justification for *why* one would want to estimate a covariance matrix from an AR process, rather than the dynamics matrix, is in order. in the end of the manuscript, you demonstrate an important application that totally justifies, but leading up to that, i was wondering.
Summary: very nice, could become new standard, provided some guidance on choosing b is provided, and demonstration that performance is robust to this choice of b, and accuracy is not so much worse than cross-validation.

Submitted by Assigned_Reviewer_21

The authors propose a novel bias-corrected estimator of covariance matrices for autocorrelated data. They provide simulated data as well as a real-world data set on brain-computer interfacing to demonstrate the superior performance of their estimator in comparison to a standard-, a shrinkage-, and the Sancetta estimator.

I believe the authors address a very interesting problem and make an important contribution. At the same time, I find the manuscript rather hard to read and the experimental result on real data not particularly convincing:

* The authors do not really introduce their notation. While most notation is obvious from its context, this makes the manuscript harder to read than it would need to be.
* I did not quite understand the heuristic fix of the Sancetta estimator in Section 2.
* Along the same lines, I would be interested in a more detailed explanation of the bias-corrected estimator in (6). As the discussion section is primarily a summary of what the authors have done, it might be shortened to have more space in Section 2?
* I am missing some of the details of the decoding procedure in Section 4. In particular, how many CSP filters were used for decoding? Which frequency band did the authors use? How did they perform cross-validation?
* The following is the primary concern I have with this manuscript: It appears to me that the authors use only two trials to estimate the CSP filters and only pick the two most discriminative CSP filters for the plots in Figures 6 and 7. This is not what one would typically do in this setting. I suspect that this choice has been motivated by highlighting the differences between the different estimators. Furthermore, it appears to me from Figure 7 that the differences in performance between the various estimators are not a result of a better estimation of the spatial filters, but rather due to a different ranking of the CSP components. One would typically not use the best two but the best six CSP filters. From my experience, it is quite likely that this set of CSP filters would include filters for left- and right sensorimotor cortex for all estimators. If so, the differences in decoding performances between the estimators are likely to be negligible. In order to be convinced that the bias-corrected estimator outperforms any other estimator, I would like to see decoding differences on CSP filters that at least focus on the same brain regions.

Typos and minor comments:
* I believe the normalization term is missing in (3)?
* Section 2: "rate of p" should be "rate of n"?
* The first sentence of the second paragraph in Section 5 is very hard to parse.
Summary: Very interesting theoretical work. The experimental results on real data, however, have been tuned to look more impressive than they would be in a realistic setting.

Submitted by Assigned_Reviewer_43

The aim of the paper is to provide an unbiased and consistent method for the accurate estimation of covariance matrices in presence of high dimensional dataset with small number of examples subject to internal autocorrelation that further reduce the effective size of the datasets. The solution is based on a state-of-the-art approach proposed by Sancetta [San08] where covariance matrix is shrinked toward a diagonal matrix with a shrinkage intensity proportional to the variance of the covariance matrix. In this framework the paper proposes an analytical estimate of shrinkage intensity incorporating a bias correction that relates the coefficient with the effective size of data.

The proposed unbiased variance estimator is an incremental work ([San08]) and it does not provide a strong theoretical novelty. However, the advantage of proposed solution is theoretically sound, and an empirical evaluation on toy examples and on a real EEG dataset shows that the proposed estimate is actually comparable to the one in the original work and it is even better in case of small high-dimensional datasets. A comparison with CV (not just the computational cost) would have been very useful.

Technically, while being relatively clear, the paper has some flaws, as many times the notation is used without any introduction of its meaning, making it sometimes difficult to follow all the formulations. Moreover, I noticed changes in the formulation along the paper (indexes inversion). It seems that X is interchangeably assumed to be organized by rows or by columns. Finally, the figures are many times difficult to understand because of missing descriptions both in the captions and in the text.
Summary: The paper proposes an incremental work with a limited originality, nevertheless, the proposed solution presents some advantages which have been proven theoretically and empirically. Technically it is well written but still needs some work to make it clearer, principally correcting some mistakes in the indexes and introducing the formal notation the first time it is used.
Author Feedback
Author rebuttal: First we would like to thank the reviewers for their detailed feedback containing very helpful suggestions and their patience with numerous glitches having occurred under deadline stress.

Second, we would like to restate the main messages which we will try to make clearer in a revision of the manuscript:
independently of the application domain
(1) neglecting autocorrelation in analytic covariance shrinkage leads to a strong bias
(2) the state-of-the-art method is still biased and highly sensitive to parameter choice; our proposed method is strictly better.
(3) our theoretical results translate to real world data.
While (1) is not new --otherwise no state-of-the-art method would exist-- it is little known and often not taken into account by practitioners, examples include [1,2,3].

Detailed discussion of main feedback:

(A) how to choose the additional parameter b? (Reviewer 13)

Figure 4 shows that this is a decisive advantage of our proposed estimator: while the Sancetta estimator is very sensitive to the number of lags b, our estimator is practically invariant as long as the autocorrelation decays almost completely within the chosen number of time lags.

(B) comparison to cross-validation, trade-off between computation time and accuracy (Reviewer 13, 43)

- We have reanalyzed the BCI data set with cross-validation and performed the statistical test: our proposed method is slightly better for a small number of trials, but the performance difference is not significant at the 5% level. We will include the results in the figures.
- In practical applications, the performance difference between cv and analytic shrinkage tends to be low: it has been observed that cross-validation is sometimes not a good predictor for BCI performance [1,5], and in ERP analysis, analytic shrinkage has become state-of-the-art [4].
- The importance of saving computation time depends on the application. For instance, very large or online applications might be very time-critical. The best response is that for many researchers and practitioners, the difference in computation time is an issue as shown by the wide usage of Ledoit-Wolf shrinkage [6].

(C) decoding process, number of trials, and number of CSP filters (Reviewer 21)

The manuscript focuses on the theoretical properties of shrinkage estimation under autocorrelation and we therefore kept the section on BCI results short. Yet we agree with one of the reviewers that the section is too short and slightly unclear w.r.t. some aspects. To clarify:
- the frequency band was optimized for each subject [7].
- for figure 5, the number of trials was varied between 2 and 20 per class. As the reviewer writes, the low number of trials for figure six has been chosen to highlight the differences between the estimators.
- The number of CSP filters was 1-3 per class, adaptively chosen by a heuristic [7]. Hence, the difference in performance is not an effect of the ranking of filters.

(D) AR(1) model and justification of the method (Reviewer 13)

Maybe the AR(1) model is a bit over-emphasized in the manuscript: the theory is more general and the estimator applicable to any autocorrelated time series. In our manuscript the AR(1) model is only chosen as the most intuitive and easy to understand example for an autocorrelated time series. In fact, if one knows that the data comes from an AR(1) model, it would be better to directly estimate the model. We will try to make this more clear in a revision of the manuscript.

References:
[1] Lotte et al., ToBE 2011 http://hal.inria.fr/docs/00/52/07/54/PDF/tbme10.pdf
[2] Samek et al. NIPS 2013
http://papers.nips.cc/paper/4922-robust-spatial-filtering-with-beta-divergence.pdf
[3] the MNE toolbox (http://martinos.org/mne/stable/generated/mne.decoding.CSP.html)
[4] Blankertz et al., Neuroimage 2010
http://eprints.pascal-network.org/archive/00008033/01/BlaLemTreHauMue10.pdf
[5] Blankertz et al., NIPS 2008
http://machinelearning.wustl.edu/mlpapers/paper_files/NIPS2007_983.pdf
[6] Ledoit and Wolf, JoMA, 2004
http://perso.ens-lyon.fr/patrick.flandrin/LedoitWolf_JMA2004.pdf
[7] Blankertz et al., IEEE Signal processing magazine 2008
http://eprints.pascal-network.org/archive/00003318/01/BlaTomLemKawMue08.pdf